# Response Strategies of UK Construction Contractors to COVID-19 in the Consideration of New Projects

Oliver Rhodes, Ali Rostami *, Atousa Khodadadyan and Sian Dunne

Department of the Built Environment, Liverpool John Moores University, Liverpool L3 5UA, UK; oliver.rhodes@bepdt.com (O.R.); a.khodadadyan@ljmu.ac.uk (A.K.); s.l.dunne@ljmu.ac.uk (S.D.)
* Correspondence: a.rostami@ljmu.ac.uk

**Abstract:** COVID-19 presented a catastrophic event, creating a unique environment and resulting in lasting repercussions globally. The construction industry has been one of the worst affected sectors relating to the public health pandemic. Challenges such as workplace closures and site cessations led to untold uncertainty, developing into contractual grievances and supply-chain disruption, amongst others. The focus of this study is to determine the response strategies of UK construction companies in the face of the COVID-19 global pandemic and the subsequent recession the UK fell into as a direct result. A literature review of previous recession responses was examined and four areas for further consideration were identified, which included contracting, risk management, cost control and finance. The study compared the previous response strategies to identify whether lessons had been learned from prior experience, or if new strategies had emerged due to the different economic and political circumstances. A qualitative methodology was adopted to provide the required depth of analysis for the research. Thirty-two participants from different size construction organisations were interviewed, which provided evidence of strategies across the four categories analysed. The results indicated that in the early stages, uncertainty around all aspects of the pandemic caused organisations to anticipate the worst financial consequences, as the scale or scope of government intervention was initially unknown. As a result, companies reacted by downsizing, halting expansion, introducing competitive pricing to ensure there were projects in the pipeline and diversification to ensure stability and survival of the company. Organisations used the pandemic as an opportunity to restructure and invested in new technology to remain competitive. Client relationships and supply-chain partnerships were deemed to be of upmost importance in resolving contracting challenges that the pandemic brought about.

**Keywords:** construction management; UK construction supply; construction business assessment and COVID-19

## 1. Introduction

The construction industry has been one of the worst affected industries directly relating to the COVID-19 pandemic [1]. Total workplace closures and site cessations were among the early signs of uncertainty and problems the industry would face, bringing untold uncertainty for an indefinite period [2]. As the pandemic materialised, initial closures developed into contractual grievances, supply-chain disruptions and other major issues hindering the sector with immeasurable costs and burden [3].

Workplace closures, a persistent factor of global pandemics, have decreased productivity. Infections resulting in the reduced supply of labour and lockdowns shutting nonessential businesses, in addition to heightened uncertainty have meant that economic decline has been unavoidable, with a large proportion of the economy coming to a halt [4]. In terms of the construction sector, private and some public sector projects ceasing to operate, project backlogs were experienced. This led to numerous other issues including termination or suspension of current contracts, and future tenders being put on hold [5]. The resultant slowdown, with delays and disruption rife within the sector, has rendered

some projects impossible to complete, leading to losses and strain on organisations within the industry [6].

In addition, the global macroenvironment changed as a result of COVID-19 in terms of aggregate demand and total supply, labour income and financial market trade [4]. Disruption to supply and demand manifest in three principal areas as a result of COVID-19: direct effects on global production, supply-chain and market disruption, and financial impact on organisations and markets [7]. The dramatic contraction caused by global lockdowns and the halting of production created uncertain timeframes and caused knock-on effects across the global economy [8]. The full impact of COVID-19 on the UK economy will not be ascertained for some time, although short term indicators can provide some evidence of the effects [9]. The initial impact was on the supply side, as factory closures in China then moved to the West, causing contractions in the macroeconomic supply. In turn, reduced output resulted in shortages and increasing global prices, a phenomenon known as 'stagflation' [10]. Demand-side responded by a reduction in interest rates, but due to an inability to find alternate supply in the short term, global shortages and inflation were observed as a result [10]. This has proven to be a vulnerability in organisation supply chains [7]. Owing to the sector's heavy reliance on overseas supply, the cessation of travel and trade ceased with the closing of borders. This has been further exacerbated by lockdowns and restrictions that halted the supply of materials for the primary, secondary and tertiary sectors. In the context of construction, companies have not been able to obtain materials to complete projects because of trade restrictions and the heavy reliance on eastern exports; additionally, as organisations have also adopted lean production methods and other efficiency gains, inventory levels are nonexistent, causing the inevitable suspension or termination of contracts unless cost-effective alternatives could be sourced [11]. Further impacts followed as closures occurred, with consumers cutting back on spending, which reduced the demand side, causing GDP to contract and causing unemployment [10].

Fiscal and monetary policies have been used on numerous occasions in an attempt to stabilise construction cycles [12], which result from the disproportionate share of corporate insolvency and individual bankruptcy in relation to the wider economy in the UK [13]. In addition, the construction industry is among the greatest contributors to GDP in the UK, attributing to the purpose of prioritisation regarding policy [14]. Adaptability and reaction to change is essential for survival and to avoid insolvency. Natural systemic fragmentation increases industry sensitivity to economic cycles and increases the rate of business failure [15], posing the question whether a high proportion of insolvencies could be prevented through the correct government policy. The construction industry is relatively labour intensive, with previous estimates valuing labour at 35–50% of total costs [16]. Therefore, as wages increase, the cost of construction inflates. In previous recessions, as monetary policy has ensured high interest rates, investment has fallen, resulting in a worsening of market conditions [16]. In recent history, recessions and high inflation have occurred together, usually combined with a high interest policy [16]. The cost of capital increases with high interest rates. As the industry has a high requirement for capital, demand decreases concomitantly [17], and as inflation increases labour costs increase, causing a fundamental problem for the industry [16]. Although the low interest rate environment that characterised the COVID-19 pandemic may have stimulated investment and growth, liquidity problems run rife, as finance is limited. Together with restive flows through the economy, there is little financial incentive for lenders to lend, thus restricting capital flows further [18]. Furthermore, construction organisations are affected by the level of external debt, demonstrating a direct relationship with the wider economy and the requirement for lending in these periods. This is a deterrent for investment and creates a vicious cycle, which causes further contraction of the economy [19].

Preventing recession is a top priority of governments and can be extremely challenging, the most important factor being economic policy [20]. Through manipulation, this attempts to force the behaviour of individuals and organisations to either save or spend, depending on interest rates and the policy introduced, essentially restarting the economy

and increasing capital flows [20,21]. The stimulus package, including a variety of measures aimed at employees and businesses, totalled 330 billion GBP over a 20-month period [22]. The stimulus package provided by the UK government to ameliorate the impact of shutdowns on organisations and employees included loans, debt purchasing and the furlough scheme. In the short term, these measures support the economy, providing jobs and keeping businesses afloat. However, issues surrounding the liabilities will continue as cost obligations will need to be paid. National debt will increase, creating the conditions for hyperinflation due to the injection of liquidity into the economy, money the government has had to borrow [22], indicating the importance of internal company response strategies for organisations to survive or thrive in economic disparity.

Cyclical fluctuations, which characterise economic growth, have an intrinsic relationship with the level of construction activity in that stage of the cycle [23]. As the nature of the business environment changes constantly, the effects on businesses also change [24]. Therefore, organisations need to adapt and respond to change, as business failures are disproportionally concentrated in these periods with little diagnosis or prescription [25]. Tansey et al. [26] suggest that in volatile market conditions, it is difficult to obtain a consistent flow of contracts and therefore companies must be dynamic and be able to react to change quickly, or risk failing. Similarly, much of the existing literature suggests the importance of knowledge banks and lessons learned from previous times of economic hardship. Yet, the shock of COVID-19 causing the downturn, the unique economic and legislative restrictions applied, having a versatile and dynamic supply chain, along with sufficient capital or work in the pipeline, has emerged as more important when defining response strategies [26,27].

This study aims to examine the most effective corporate response strategies of UK construction companies to mitigate the economic effects of a pandemic. The objectives are to understand the effects of recession on UK construction organisations and what strategies have been previously employed in this regard, which are then used to compare with the response strategies immediately following the COVID-19 pandemic. Analysis of these strategies is used to recommend future considerations to prevent issues arising from downturns in the business cycle.

Recessionary periods are inevitable and are indiscriminate toward their victims. With many of the 'larger' construction firms appearing to remain unscathed during these periods, it seems appropriate to investigate the challenges of different sized UK construction firms. To understand the full extent that a recessionary period has on construction, the appropriability of research design is of the essence. Research can be challenging due to the construction industry's dynamic nature [28]. Therefore, correct research design of paramount importance.

The scope of the research was formulated off the basis presented from past research focus and the current organisational landscape that construction companies are presented with. Table 1 presents the focus of historical research on response strategies of construction companies, which has provided a foundation to narrow the field of view for the current research. From this research, the categorisation of response strategies was often found with blurred lines throughout the process, with overlap and variation occurring among published research reports. To combat this, popular and generic categories were selected, which may incorporate other categories from various literature. The categories selected are as follows: contracting, risk management, cost control and financial. Conjointly, categories selected are associated with KPI's during construction operations, and are therefore directly correlated to response strategies in adverse environmental conditions.

A qualitative methodology was selected to provide the required depth of analysis of the research. An inductive approach was undertaken to allow the researcher to explore the topic at a satisfactory depth and to capture the participants' perceptions of the threats faced by their firms. Organisations of different sizes across several sectors were chosen so that the results would be superior to a larger analytic nomological sample. The open-ended nature of the questions, attention to attitudes and the incorporation of human experience justify

this methodology [38]. Furthermore, the qualitative nature encourages reliability and validity as all participants were anonymous, while the environment allowed the researcher to obtain the actual experience and responses of the organisations they employed within this period. Other studies, such as Danforth et al. [35], conducted their research through interviews, for the reasons stated above, which has assisted in building the research area due to its strengths, such as flexibility and exploration value, which some quantitative methods do not provide [35].

**Table 1.** Response strategies.

| Author | Response Strategies | | | | | | | |
|---|---|---|---|---|---|---|---|---|
| | Contracting | Financial | Risk Management | Cost Control | HR | Investment | Strategic | Operations |
| Lansey [24] | | | X | | | | X | X |
| Hillebrandt et al. [27] | | X | | X | | | X | |
| Pearce and Michael [25] | | X | X | X | | | X | |
| Ocal et al. [29] | X | | X | | | | X | X |
| Lim et al. [30] | X | X | | X | | | | |
| De Waal and Mollema [31] | | X | | X | | | | X |
| Li and Ling [32] | | | | X | | X | | |
| Honek et al. [33] | X | | | X | | | | |
| Jung et al. [34] | X | | | | | | X | |
| Tansey et al. [26] | | | | X | X | | | |
| Ruddock et al. [23] | | X | | | X | | | X |
| Danforth et al. [35] | X | X | X | X | X | X | | |
| Frick [36] | | X | | X | | X | | |
| Raoufi and Fayek [37] | X | X | X | | X | | | |

## 2. Methodology

This research is designed based on a qualitative method to identify the key response strategies of UK construction contractors to COVID-19. The findings related are currently being reviewed and extended for the purposes of developing a framework that might be deployed in the future by construction practitioners. Based on the extant pertinent literature (see Table 1), response strategies were considered to identify the key factors faced by contractors in new construction projects. This exercise provided the opportunity to refine and augment the content of the literature review and, moreover, to probe some of the key issues faced by practitioners on the ground. The exercise also added breadth to the research and formed the basis for the content of semi structured interviews. Qualitative semi structured interviews were conducted during the data collection phase of study. All the participants were current professionals working within the UK construction industry. To ensure the quality of the information collected, participants were provided with an outline of topics to be discussed in advance. Each participant performed one of the following professional roles: senior project manager, project director or project coordinator.

Interviews were conducted one-to-one over Zoom and Teams video conferences due to the restrictions in place from the COVID-19 pandemic, and were recorded to allow detailed analysis, to ensure reliability and accuracy. Recording also helped in the transcribing process and allowed the interviewer to give the participants full attention. Participants were recruited through email and professional social networks, ensuring the participants had the attributes relevant to the study. Stratified sampling was used in the recruitment process. Through population subgroups (strata), this ensured that appropriate participants could be selected based on their attributes/group membership [39]. In this case, the requirements were employment in the UK construction industry and having at least 5 years work experience. This ensured that respondents understood the effects that COVID-19 has had on their company and the wider industry, including the difference between the pre- and post-COVID-19 periods. All participants remained anonymous in the study, preventing any liability or subsequent issues arising by divulging information. Furthermore, questions were tailored to avoid participants providing sensitive or company-specific information, which may inhibit themselves or the organisation. The interview questions and related

responses were split into four principal sections, derived from the principal literature research, as follows: contracting, risk management, cost control and financial. The full list of questions is detailed in Appendix A.

## 3. Results

Pope et al. [40] indicate the importance of examination and analysis of raw data, allowing any patterns and themes evident in the data to define the relationships between multiple factors or cohorts. As the method of enquiry was by interview, the qualitative data gathered and analysed is presented in this section. Table 2 displays the total responses by category from the whole sample, with the overall total being a frequency of 177. The majority of responses involved cost control, with 33.3% ($n = 59$) of the total number. These were divided into three subcategories: (1) cost reduction, (2) efficiency, and (3) human resources. There were 27 different strategies representing the cost control responses, with the frequency representing the number of samples utilising the strategy. Contracting-related responses represented the second most popular response category, generating 23 different responses and comprising 24.3% ($n = 43$) of the total responses. The strategies employed were divided into six different categories as follows: (1) diversification, (2) client relationships, (3) enterprise tactics, (4) bidding, (5) subcontractor relationships, and (6) marketing. Following on, risk management strategies were the third most popular category comprising 21.5% ($n = 38$) of the total responses with 18 different individual strategies. The responses were classified into five different subcategories, as follows: (1) risk identification/mitigation, (2) unfamiliar risk, (3) recovery risk, (4) project risk, and (5) enterprise risk. Lastly, financially related responses comprised 22 different strategies, representing 20.9% ($n = 37$) of the total strategies employed by the sample, and, therefore, a minority of the total results. The responses were allocated into three subcategories as follows: (1) profit and cashflow, (2) investment, and (3) payment terms.

**Table 2.** Total responses by category.

| Response | Frequency | Percentage |
| --- | --- | --- |
| Cost control | 59 | 33.3 |
| Contracting | 43 | 24.3 |
| Risk management | 38 | 21.5 |
| Financial | 37 | 20.9 |
| Total | 177 | 100 |

## 4. Discussion

The concept of organisational and behavioural changes dependent upon the position within the economic cycle has been investigated and accepted by many authors previously [23,26,27,30,35]. The rarity of a global pandemic that changes everyday life means that its effects have not been explored in the way that other recessions have. The last event similar to the period in question was the global Spanish flu epidemic of 1918, but there were considerable differences with government regulation, the research of the pandemic and the availability of the data. Therefore, there is a lack of knowledge regarding the response of construction companies in this environment, indicating that research is required.

Previously conducted studies found that responses were split into subsections, although they were categorised slightly differently. Sections included contracting, financial, risk management, strategy, investment, human resources and cost control, which were among the subsections presented by previous authors [27,30,31,35]. These have been analysed, and based on the answers of participants, were classified into four response categories in the present study: cost control, contracting, risk management, and financial. Furthermore, to expand the knowledge base, such as in Danforth et al. [35] each area was further categorised into the subsections presented in the results section. Not only was the intention to expand the body of knowledge, but it also allowed patterns and comparisons in behaviour with previous recessions to be recorded. This should aid the practical application

of the research and provide construction companies with clear guidance, if an event with similar characteristics or magnitude were to occur again.

## 5. Main Findings

The study produced 91 unique strategies in the four categories studied. These categories were also highlighted by Lim et al. [30] and Danforth et al. [35]. The subcategories were categorised based on the participant responses and were refined accordingly. The number of unique responses indicates a situation in which UK construction companies had to react to remain competitive. The individual responses from the organisations are mainly from a reactionary view, the majority occurring after the event. This indicates the spontaneous nature of the response to the pandemic and the uncertainty around it, with limited knowledge and time to prepare. As legal restrictions were implemented across the UK, one of the main findings was the organisational changes companies had to endure to continue operations whilst adhering to the restrictions. The results demonstrated that 100% of the samples experienced an increase in virtual presence, as working from home was the main change. Not only did this occur during the restrictions, but many companies also opted to continue remote working in some form, benefitting from the cost saving it provided. This was a key difference from previous studies where organisational changes stemmed from the hardship imposed by recessions, rather than legislation imposed by government [30,34].

The restrictions also caused a complete closure of sites and 'nonessential' businesses, a period of three months during which many projects were paused. This resulted in responses in all four categories, although the financial strains caused by other recessions due to increased competition and reduction in demand were not observed [23,36]. As government schemes, such as furlough and loans were rolled out, companies were able to cover liabilities and pay employees, reducing the strain placed on their organisations. Although respondents also indicated that projects were paused, contractors were not receiving payments and subcontractors were not paid, causing a ripple effect all the way through the supply chain. Furthermore, a significant finding was uncovered, caused by the expected lag of the real effects of the recession. As all work was paused, backlogs of projects have occurred, pushing back future projects which could affect corporate viability; as government schemes end, costs are likely to increase in each organisation, adding to the potential termination of future projects and the possibility of reduced demand in the future [37]. Although the effects of previous recessions have been endured, several participants voiced concerns about the future.

Cost control was the most frequently stated response, representing a third of the total responses. This reveals the uncertainty of the situation, with companies opting to reduce costs as there was no timescale for the pandemic and no indication of how long restrictions may last. This also indicates the importance of cashflow and the need for companies to stay afloat in an adverse economic climate [13,41]. The least popular response was financially related. This could be attributed to the pausing of the industry and the economic support given by the government. As restrictions were lifted and sites reopened, many projects have continued, leading to the resumption of payments, therefore reducing the number of financial actions and responses.

### 5.1. Cost-Related Responses

A cost control strategy involves reducing both fixed and variable costs of a business through a variety of means. In the context of the current research, this ensures financial stability, with cashflow remaining positive in uncertain times. The strategies have developed since first being recognised by Hillebrandt [27], who recommended cost control with a strategy of reducing permanent employee wage growth due to the impact of lower workload and a smaller number of required staff. As confirmed by the results of the present study, organisations have incorporated a much wider view of different factors in relation to cost control from previous studies, suggesting that there is growing competition in the

business environment. The cost control responses represented a third of the total results (33.3%), creating 27 unique strategies, a clear majority. The strategies were divided into three subsections, consisting of: (1) cost reduction, (2) efficiency, and (3) human resources (HR) (Table 3).

**Table 3.** Cost-related responses.

| Subcategory | Response |
|---|---|
| Cost reduction | Use of government furlough scheme to pay employees |
| | Continue with remote work to reduce costs |
| | Reducing number or moving premises |
| | Reduce overheads |
| | Expansion put on hold |
| | Policy and mitigation charged to client |
| | Reduced number of contractors used |
| | Debt reduction strategies employed |
| | Outsourcing admin overseas |
| | Closing sister company |
| | Stop dividend payments |
| Efficiency | Diversification of employee skills (cross training) |
| | Decreased efficiency resulting from decreased workload |
| | Employment of new technology |
| | Increased capacity utilization |
| | Increased workload post-period |
| | Use of cash reserve |
| | Use of resources from larger subcontractor |
| | Increasing audit |
| | Introduction of referral bonus for recommending new employees |
| | Grouping organizational sectors together |
| Human resources | Stop pay reviews |
| | Freeze recruitment |
| | Employee redundancies |
| | Working overtime |
| | Increased employee training |
| | Bonuses stopped or reduced |

The use of the government furlough scheme to pay employees was the most popular response, with 100% of the sample indicating that their organisation took advantage of it. This comes as no surprise, considering that restrictions were applied for 20 months, which ensured stability and survival. According to Lim et al. [30], proper financial management is a key to the survival of construction firms. As periods of lockdown resulted in zero income, companies had less capital to cover overheads, demonstrating the importance of this response strategy. In these periods, raising profit is challenging, therefore by exploiting cost reduction, firms can utilise it as a measure to prevent losses [23], rather than as a strategy to increase profits. Although government policy aided in mitigating costs, as there was no ability to estimate the loss in opportunity cost, the respondents indicated that the company needed to be in the best financial position because of uncertainties surrounding the pandemic. The results of Lim et al. [30] mirrored these responses, as the eight-year recession described in their study had similar uncertainty variables.

Similar responses included the halting of pay reviews (100%), the freezing of recruitment (80%), and conducting employee redundancies (60%), responses consistent with previous studies. Lim et al. [30] observed scores of 91% for the same metrics when Asian contractors were faced with a prolonged recession. These responses are key components of Hillebrandt's definition of cost control retorts [27], essential for reducing overheads resulting from disruption caused by recessions. Moving or reducing the size of premises and preventing expansion were common within the respondents to prevent adverse financial conditions, as some organisations were not eligible for mortgage or lease holidays. Many experienced this as a progressive strategy rather than a reactionary measure, executing

the strategy before it was required to protect cashflow. Remote work was a prominent response, with 80% of respondents utilising this policy. This was a compulsory part of the initial lockdown restrictions, yet many organisations chose to continue because of the cost reduction, with little impact on efficiency. Many respondents believe this will become part of the working norm going forward, because of the cost benefits and the lack of impact on quality or efficiency of operations.

In the second subsection, represented by efficiency-based responses, diversification of employee skills was the most prominent response with 80% of respondents indicating its implementation. Lansey [24] expressed the importance of creating systems that develop and enhance the diversification of skills in an environment characterised by competitive change. According to the participants, this was to assist in retaining efficiency when moving to remote working, but also to counteract any redundancies and employees' sickness due to the virus. Organisations considered this period as a window to restructure and change the working environment, with 60% of respondents indicating that the organisation invested in new technology or operating systems. This also was due to changes in the working environment and the need to remain competitive in these uncertain times. This is consistent with the observations of Lim et al. [30], who indicated that in prolonged recessions, companies used the opportunity to restructure and invest, exploring other business avenues to remain competitive. Despite this, some respondents indicated that the investment was minimal, as cash and liquidity was essential for contingency, acting as a buffer should the company face tough times, generally following the mantra that cash is king.

Efficiency from workload was another theme emerging from this subsection. In total, 60% indicated that efficiency decreased due to the decrease in workload, a natural consequence of there being zero output due to lockdown restrictions, with efficiency naturally decreasing. Following from that, 40% of respondents indicated that an increase in workload post-lockdown was observed due to the backlog of projects, with organisations increasing workload to regain some of the lost efficiency. It is more than likely that the same participants also indicated that overtime was worked because of this, as reflected in the HR section of the cost control responses. Capacity utilisation (CU) and increased audit were two other strategies companies used to target efficiency, at a rate of 40% and 20%, respectively. Using CU, costs are more efficiently allocated, reducing the cost per unit, as overall cost is usually reduced with companies moving premises and cross-training employees. Such auditing will reduce wastage, among other factors, and in addition to ensuring efficient spending, employee performance will refine cost data and improve overall company performance.

The final subsection is the HR category, which had the least number of unique responses. The most popular response was the cessation of pay reviews (100%); this reduces any unnecessary increase in costs over the period, increasing the likelihood that the organisation will maintain financial stability. Freezing recruitment (80%), making redundancies (60%) and stopping bonuses (20%), combined with the pay review action, also reduces the fixed and variable expenses of the organisation. This is consistent with Tansey et al. [26], who indicated that a combination of a freeze of salaries and making redundancies were the most popular responses, as cost leadership was a popular strategy adopted from the Porter's 5 Forces model for contractors facing a recession. These strategies were apparent in the research of Danforth et al. [35], although the proportion of companies in the present study utilizing the strategy was higher, suggesting their need was greater. This could be attributed to the uncertainty and spontaneous nature of the event compared with previous recessions [35], where organisations had no legal restrictions preventing operations in combination with the experience of past recessions in which there was more certainty about which strategies were successful. Therefore, by preparing for the worst, organisations have been able to thrive if the conditions were in fact not as bad as they had initially appeared, or after improvements, for example, following the lifting of restrictions post-lockdown. Li and Ling [32] observed that management practices such as cost reductions in

operations combined with administrative activities are not significantly correlated with profitability. Therefore, organisations should exploit other types of strategies to increase their profitability.

### 5.2. Contracting-Related Responses

Contracting-related responses are actions undertaken to obtain or continue work and maintain a company's financial position [27]. Obtaining work was less apparent than in previous recessions due to the backlog of projects accumulated because of government restrictions; therefore, in this section the study concentrated on how organisations implemented competitive strategies throughout and following the period. Despite this, large organisations always plan ahead and therefore bid for future projects to place in the pipeline. One therefore may argue that Hillebrandts' definition [27] is accurate for the present study. This response was popular among the participants, splitting into six subcategories based on the participant responses, creating 24 unique strategies, and representing 24.3% of all responses in the study. The section was divided into the following subcategories: (1) diversification, (2) client relationships, (3) enterprise tactics, (4) bidding, (5) subcontractor relationships, and (6) marketing (Table 4).

**Table 4.** Contracting-related responses.

| Subcategory | Response |
| --- | --- |
| Diversification | Increased virtual presence |
| | Diversification of markets during period |
| | Diversification of geographic location |
| | Diversification of markets post-period |
| | Vertical diversification |
| | Pursue new project sizes |
| Client relationships | Use of repeat/familiar clients |
| | Importance of client relationships |
| | Working for new clients |
| | Clients more demanding |
| | Reduced communication with clients during period |
| | Stop working private contracts |
| Enterprise tactics | Demand reduced during period |
| | Increased competitiveness to increase demand |
| | Maintain business plan despite economic climate |
| Bidding | Increased competition |
| | Not bidding for low value projects |
| Subcontractor relationships | Importance of relationships |
| | Use of new subcontractors |
| | Use of new suppliers |
| Marketing | New marketing campaigns |
| | Increased social media presence |
| | Marketing on motor vehicles |

Diversification and client relationships were the most popular response categories, with 17 individual responses each. According to Jung et al. [34], diversification is a strategy that increases profitability through rapid movement into different markets and products. This was the most popular strategy reported by Danforth et al. [35] and Tansey et al. [26], indicating that it was effective in past recessions, as it was used in the face of the current recession. Increased virtual presence as mentioned in the last section was the most popular, with 100% of the respondents indicating its utilisation, which is likely to change general working life. Diversification of markets during the period and diversification of geographic location both represented 40% of responses. Companies were able to shift resources to these markets and other locations, indicating flexibility within the corporate structure. As some of the large organisations have decentralised organisational structures with adequate resources and capability, this has allowed them to diversify whilst increasing profitability.

This is in line with results from Danforth's [35] and Lim's [30] research. Jung et al. [34] also found that geographic diversification was effective in the award of new contracts, with 40% of respondents in the present study indicating that their company was involved in the strategy.

Client relationships were another popular category, with six strategies employed. The importance of client relationships and the use of repeat/familiar clients were the two most popular responses, with a 60% utilisation rate. Wong and Logcher [12] suggested that reputation and relationship quality play significant roles in allowing companies to survive or thrive in a recession, especially applicable in the private sector [12]. Lansey [24] contradicted the statement, expressing the need to develop new relationships in troubled times. The present study indicates that both were utilized; working with new clients was utilized by 40% of the respondents. Whether the relationships are established or fresh, firms with better relationships weather downturns better and are more likely to thrive, according to Hillebrandt [27]. The participants representing smaller firms indicated their gratitude to some of their clients, as their relationship may have been crucial to survival after the pandemic period, after the government incentives ended. There was little indication of the strains on relationships as noted in Danforth et al. [35], as hardships experienced during the period were not as prominent due to government aid. However, one participant indicated that clients were more demanding over this period. This could be attributed to strains that such clients were experiencing themselves, providing contractors less flexibility within the constraints of the contract. Finally, due to the backlog of projects in the pipeline and the number of public sector contracts awarded to them, one respondent indicated they had halted working on private contracts. There tends to be greater availability of government contracts during recessions to stimulate the economy. These are attractive to construction firms because of the reduced construction costs associated with them [35].

The previously published literature relating to construction demand in a recession suggests that a slump in demand is experienced within downturn periods [24,27,30,32,35]. Sixty percent of participants agreed with this, although their experience differed slightly. As companies now work so far ahead organising contracts for the future, pausing projects caused significant disruption and uncertainty. As the projects were delayed, resources from companies were reallocated to execute the current contracts. As some organisations were not actively searching for new contracts during the months of lockdown, due to company closures, few or no new projects were available, with demand levels recovering after the lifting of most restrictions. This resulted in increased competitiveness after the pandemic period according to 40% of the participants, as organisations moved from being in a passive state to actively pursuing new projects, as many predicted future struggles after the period. This was further represented in the bidding section with 60% indicating an increase in competition. As public sector contracts become more available, the number of organisations tendering for projects increased because of the cost and profit benefits associated with them. This process reduced the profit potential, with one participant indicating their reluctance to bid for low-value contracts. This could be the reason for the high subscription to diversification strategies and new technology, to identify new and diverse ways to lower project costs, winning contracts and retaining or increasing margins. Two respondents indicated that there had been no change to the business plan during this time, a result of booming demand with many projects lined up. This action is supported by Danforth et al. [35], as diverse business plans prepare organisations for success, regardless of the economic conditions in which they operate.

The subcontractor relationships subcategory was represented by only three unique strategies, yet participants discussed their importance and implications. Firstly, the importance of relationships was essential for many of participants. As the cashflow of several organisations fluctuated, such relationships allowed increased payment terms and ensured retention of good rates. This was essential for both contractors and subcontractors who both felt the effects of the recession during this period. Respondents indicated the engagement of new subcontractors. As competitiveness increased across multiple industries, the

domino effect from construction companies trying to apply better rates to win contracts influenced subcontractors. Therefore, new relationships and agreements were formed with subcontractors offering better rates, ultimately reducing construction costs for contractors. Furthermore, as the insolvency of subcontractors occurred, new subcontractors had to be recruited to enable the completion of contracts, as evidenced by the participants of the study. This has been noted in previous studies, such as Lim et al. [30] and Li and Ling [32]. For similar competitive reasons, new suppliers were also used, with 45% of the interviewees indicating this. Furthermore, as supply chains were so heavily disrupted, imports, especially from China and India, were not readily available. Therefore, diversification of suppliers had to be conducted in order to complete contracts.

The final subsection relates to marketing, which was represented by only three unique strategies, each having eight responses, which were the least utilised responses. Roberts [42] has indicated that marketing should be intensified during a recession to exploit the competitive advantages of the organisation. Increased social media presence, marketing the company on motor vehicles, and creation of new marketing campaigns were the responses initiated during the period. The lack of marketing campaigns resulted from the backlog of projects and a focus on other functional areas of the organisation according to some participants, whilst others indicated that the capital from less critical budgets (including marketing) were cut and used in other functional areas of the business to ensure liquidity in cash-stricken periods. Hillebrandt [27] has indicated that marketing strategies should be altered in these periods, but should not be decreased, in fact, the opposite.

*5.3. Risk-Management-Related Responses*

The third section relates to risk-management responses, capturing the risks that organisations have faced, and the strategies employed to mitigate such risks. In previous studies, there has been clear neglect of this aspect when discussing the strategies employed by organisations. Only research studies by Pearce and Michael [25], Ruddock et al. [23] and Danforth et al. [35] describe in detail the strategies undertaken in this section and the importance of this category in the face of a recession. Within this section, five subcategories were identified based on answers from the participants, as follows: (1) risk identification/mitigation, (2) unfamiliar risks, (3) recovery risks, (4) project risks, and (5) enterprise risk (Table 5).

Risk identification/mitigation was the first subcategory identified, of which there were five unique strategies, which were used to protect the companies against risks either created or present during the period. All respondents indicated that COVID-19 clauses had been included in contracts. For example, the extension of time, maximum site presence and other issues that could arise were also considered and inserted. Participants indicated that prior to resuming any work after lockdowns, meetings with all parties were conducted to consider such events, as it was in the interests of all parties to work together and cooperate to complete each project. This contrasts with previous studies which indicated that contracts can be terminated as a result of excessive delay. During the problems created by COVID-19, there was little indication from the participants that this had occurred [37]. However, one participant indicated that force majeure had been utilised, as the project pre-COVID-19 was in an unsatisfactory state. Because of the disruption caused by the pandemic, the parties saw no way of completing the project. Participants identified the ways that their organisations had mitigated risks, representing 20% of the responses. COVID-19 risk was passed onto clients in one circumstance, which had been agreed when revising the contract post-lockdown, putting the organisation in a more favourable financial position. This is rarely observed in contractual negotiations and not recorded in previous research but can be agreed by mutual consent, so that a contract can be completed in usual times. Specialisation in certain sectors, rather than diversification was used by one participant. By providing overwhelming resource in a particular sector, the organisation was able to gain a competitive advantage through refining their core competence and by their relationship

with clients. This can reduce opportunity during boom periods as competencies in other areas may become obsolete due to the narrow view.

**Table 5.** Risk-management-related responses.

| Subcategory | Response |
| --- | --- |
| Risk identification/mitigation | Insertion of COVID-19 clauses into contracts |
| | COVID-19 regulation from governing body |
| | COVID-19 risk passed onto clients |
| | Specialisation to reduce risk |
| | Use of force majeure |
| Unfamiliar risk | COVID-19-related risk |
| | Increased contingency planning |
| | Diversification of new market risks |
| Recovery risk | Specialisation limits options post-period |
| Project risk | Changes in policy affecting site operation |
| | Reduction in site presence |
| | Riskier project undertaken during this period |
| | Project documents contain more unknowns |
| | Competitors bidding too low |
| | Lump sum risk in new contracts |
| Enterprise risk | New technology risk |
| | Rapid growth post-lockdown and closures |
| | New regulations more challenging |

The unknown risk was the next subsection. COVID-19-related risks were unfamiliar and changed the operation in many aspects comprehensively, affecting 100% of participants of the study. As legislation changed throughout the pandemic, organisations had to adapt and be flexible, adhering to new regulations whilst trying to remain competitive and profitable. According to Pearce and Michael [25], firms should invest to bolster their position in uncertain and adverse economic times. This was observed to a lesser extent, but through legal advice, diversification and exploration of new options, organisations were able to identify cost effective ways to mitigate unknown risks. Extensive contingency planning was undertaken by 78% of participants, likely to be representing the same companies already invested in diversification and other investment or expansion strategies. As organisations undertake such strategies, new risks emerge. Therefore, comprehensive contingencies are essential for any business plan to prevent losing a competitive position and putting the future of the organisation in question. Seven participants indicated the risk of diversification separately. These participants had contrasting views of the financial and competitive position of the organisation. One indicated that the move was rushed due to the unforeseen circumstances, because their competitive position had not been damaged. Conversely, the other respondents indicated that the move was premeditated and occurred at the right time, as the combined organisation already was making the move prior to the pandemic and any uncertain times that may follow.

Project risk was the most popular subcategory, representing six responses. Of the participants, 60% indicated that policy changes had been introduced that affected site operation, attributed to legislative changes brought about by COVID-19 to ensure site safety and that social distance measures were being used. Participants explained that due to the additional restrictions, work typically slowed because of manpower shortages due to sickness, isolation and restrictions in labour-intensive workplaces. Raoufi and Fayek [37] presented the additional bureaucratic stages that ensured the safety and well-being of employees, also indicated by respondents, slowing projects, and altering site operation. Concerning site presence, 45% percent specified that there was a reduction. The reduction was not 100% in these latter two strategies because of the line of work of some participants, as some operate on the consulting side of the industry, therefore having little or no presence on site. Despite this, all participants expressed how the pandemic altered normal operations.

Riskier projects were undertaken during this period according to 40% of the participants, attributable to uncertainty during the period. Organisations wanted to ensure there was work in the future, as there was no certainty about the duration or severity of the recession, especially after the government stimulus schemes ended. These organisations took the view that riskier work was better than no work. These projects represented additional unknowns according to the study, attributable to riskiness. This increased the difficulty in the identification and mitigation process, with contingencies hard to formulate as a result, consistent with the results from Danforth et al. [35]. Competitors that bid too low and the lump-sum risk of new contracts were identified by 20% of the sample. These are typical behaviours observed in past research, as firms wanted to ensure that cash was moving through the company, even where margins were little or nonexistent. However, as previously expressed, several organisations had projects in the pipeline and backlogs, so new projects were not required. As a result, organisations saw no need to enter projects with little profit as they viewed them as a waste of time and resources, able to be spent better elsewhere (opportunity cost).

Enterprise risk was also considered, with three individual strategies that were employed. The risk of new technology was most mentioned, with a 60% response rate, with the same three participants whose companies invested in that technology. The risks related to the cost benefit analysis of such investment, because in troubled times there is a risk that any investment may not pay off, causing an organisation to be placed in a financially adverse position and the whole organisation at risk. Respondents indicated that new regulations were more challenging, with a 40% response rate. This is due to the unfamiliarity of it, increasing the time and cost as working practices become less efficient. Quality assurance was also employed to ensure that work was 'right first time', according to the participants. This further increased the time and cost of operations. This contrasts with the study by Danforth et al. [35] in this respect, as they indicated that manpower shortages and reduced budgets increased the difficulty in contracting new work, related to new regulations as quality controls were employed. In the present study, quality assurance was employed as organisations had no time or spare budget for work to be completed incorrectly, according to participants. Rapid growth post-lockdown occurred according to 40% of participants. This is typical following a recessionary period, especially considering that the downturn resulted from closures rather than an economic shock. With the aid of stimulus packages, a restart of the economy was experienced, allowing organisations to make up for lost time. Jung et al. [34] indicated that firms should balance between extending business opportunity and securing financial risk, depending on market conditions. Too great an expansion can cause resources to be too widely spread, while if liabilities are due before an organisation has collected its payments, the organisation can be placed in an adverse financial position.

### 5.4. Financially Related Responses

The final section related to financially related responses which, according to Hillebrandt [27] have a greater effect on the balance sheet and financial stability. Danforth et al. [35] went further, suggesting that companies could perform actions to improve their financial position in preparation for and during a recession. The list of responses compared with past studies is more extensive. This could be attributed to the drastic changes in regulation, legislation and the economic environment, which is unique. The responses were divided into three subcategories, as follows: (1) profit and cashflow, (2) investment, and (3) payment terms (Table 6). These generated 22 individual responses, 20.9% of the total responses collected in the study. The reason is that there was sparser distribution of responses in this category, due to a comparatively higher number of individual responses than in other categories, thus representing a higher proportion of the overall study. This indicates that organisations have followed different strategies regarding financial responses during the period analysed.

**Table 6.** Financially related responses.

| Subcategory | Response |
| --- | --- |
| Profit and cashflow | Careful monitoring of cashflow |
| | Large fluctuations in cashflow |
| | Perform little or no profitable work |
| | Profit margins retained |
| | Profit margins decreased |
| | Profit margins increased resulting from increased workload |
| | Rigorous budgeting |
| | Increased rates |
| | Specialisation limits options post-period |
| | Noncontract work completed |
| | Implementing minimum margins |
| Investment | Investment in diversification |
| | Cost of new technology |
| | Investing in rebranding |
| | Preventing employees purchasing shares |
| Payment terms | COVID-19-related payment events |
| | Increased retention |
| | Longer payment terms |
| | Fees retained |
| | Fees diminished |
| | Payment time decreased |
| | Governing body sped up payment terms |

The first subcategory was related to profit and cash flow, providing an understanding of how the strategies used by different firms ensured competitiveness, maintaining a positive balance sheet. Careful monitoring of cashflow was the most popular response in this subcategory, with 80% of the respondents agreeing. This is not a surprise, because in turbulent times this would be a natural response, to ensure that the company is able to pay any liabilities and remain afloat. There were indicators or triggers within the internal processes of firms to identify if cashflow went below a certain threshold, with management meetings and other processes occurring to ensure company survival. Large fluctuations in cashflow were observed (60%), with one participant indicating that they had come close to the threshold during the beginning of the period, just before the announcement of the government stimulus package, whilst organisations still paid overheads and liabilities. A key strategy described in previous studies was the performance of work with little or no profit (40%) [30]. This can be attributed to increased competition and the reduction in fees as a result. As demand falls, particularly in an uncertain economic climate, some firms (usually smaller ones) enter such contracts to ensure cash is running through the business, allowing them to pay any liabilities with such capital. The participants who indicated the use of this strategy expressed that their use was minimal and only in smaller projects, and that the organisations were likely to attempt to recuperate costs through variations and other contractual clauses.

The results show clear differences in margins, with one participant in each stating that margins were increased, maintained or decreased. Some respondents were not included due to their lack of knowledge on their company's margins, and so declined to answer to prevent false reporting. These margins are systematically linked to the other responses in the study, for example, one respondent indicating a decrease in margins resulted from an increase in competition for projects, linked to a diversification strategy, as they felt they needed to offer lower margins in new sectors to win projects, ultimately leading to increased workload, increasing revenues and profits as a result. On the other hand, where margins had been maintained, the reasons given were due to company policy and a minimum margin threshold. That particular organisation also used the same business plan prior to the period with little or no general changes in the organisation. The other strategies in the subcategory are typical and have been implemented as a strategy to meet organisational

objectives in the period. The only other strategy of significance was the performance of noncontractual work. This resulted in smaller jobs, providing the business with a fast influx of cash relative to contractual work, improving cashflow and their competitive position as a result. According to Tansey et al. [26], such a strategy increases their position in terms of competition, allowing an organisation to be more flexible in contract particulars, increasing the likelihood of winning new contracts in the future.

Investment in technology and diversification were the two most popular investment strategies over the period analysed, with a 54% response rate. Organisations regarded these two as having the lowest risk for increasing their competitive advantage, with organisational competency to increase profitability. According to Jung et al. [34], differentiation strategies providing innovative products and services with high quality inputs are directly related to profitability, hence the importance of the investment. One participant stated that their organisation prevented employees from buying shares. They did not provide reasoning, but is likely to stem from preventing a reduction in control in these uncertain times, which could lead to backlash from corporate decisions or increases in dividend payments causing an increase in costs.

The COVID-19 payments were the most common response, with an 80% response rate. This is unsurprising as contractors will have recuperated costs for which they were not liable, related to COVID-19. Contract particulars will have been negotiated and discussed at the restart following lockdown. Longer payment terms were also negotiated as a result, according to 40% of participants, easing the requirement for capital in such times, ensuring that the contract could be completed correctly and through cooperation, favouring both parties. Two of the final strategies in this category relate to fees. One participant indicated that fees were retained while one stated that they had diminished. The other participants did not comment on this aspect. A reduction in fees was not surprising in this period, but the participant indicated that they bounced back quickly, contrasting with Danforth et al. [35], who indicated that the recovery in fees was slow. The retained fees are consistent with other results in the present study, as one company did not change its margins, fees or business plan, indicating that the recession caused by the pandemic had little or no effect on the organisation.

## 6. Conclusions

This research explored the response strategies of UK construction companies in the face of a global pandemic and, importantly, the response of organisations facing the greatest reduction in GDP in recent history and the subsequent recession. The research compared the response with strategies previously employed during past recessions to explore the principal differences, identifying whether lessons had been learned from prior experience, or if new strategies had emerged due to the different economic and political circumstances.

The study found that in the early stages, uncertainty around all aspects of the pandemic caused organisations to anticipate the worst financial consequences, as the scale or scope of government intervention was initially unknown. As a result, companies reacted by downsizing, halting expansion, introducing competitive pricing to ensure there were projects in the pipeline and diversification to ensure stability and survival of the company. Following the introduction of government stimulus packages and reopening after the lockdowns, participants acknowledged that the implications for their companies were less severe than had been initially feared, although changes to operations had to be implemented to comply with restrictions such as social distancing. Almost universally, for example, remote working was implemented. Thus, the research investigated how construction companies react in the face of a global pandemic in comparison with past recessions and economic shocks. The findings of the study provide an understanding of: (1) the scale and scope of the economic shock caused by the pandemic and the subsequent recovery phase, (2) the effectiveness of the response of construction companies, (3) the real effects of the pandemic following the end of various government schemes, and, finally, (4) the breakdown of response strategies implemented due to the pandemic.

The results of this study suggest several avenues for development in future research. Firstly, documentation of the full impact of the pandemic on different companies in the sector and contrasting it with previous recessions would allow organisations to make sound corporate choices during future economic shocks and downturns. Secondly, additional investigation could explore the effects on different sizes of firms, how their actions differed and the results of such actions. Finally, a mixed methods approach including quantitative methods with respondents from more diverse organisations, would allow holistic analysis of the effects of the pandemic and identification of the successful actions employed by companies.

**Author Contributions:** Literature review, O.R. and A.K.; data collection, O.R. and A.K.; methodology, O.R., A.R. and A.K.; writing—original draft preparation, O.R.; writing—review and editing, A.R. and S.D.; formal data analysis, A.K.; supervision, A.R.; proof reading, S.D. All authors have read and agreed to the published version of the manuscript.

**Funding:** This research received no external funding.

**Institutional Review Board Statement:** Not applicable.

**Informed Consent Statement:** Informed consent was obtained from all subjects involved in the study.

**Data Availability Statement:** The data supporting reported results can be found at the LJMU Research Data Repository on an Open Access basis. https://opendata.ljmu.ac.uk/ (accessed on 9 May 2022).

**Conflicts of Interest:** The authors declare no conflict of interest.

## Appendix A

### Interview questions

1.  Since the start of COVID-19, the lockdowns and the economic downturn resulting, have you noticed any threats which have arisen to the company in any aspect?

### Contracting-related responses

1.  Has there been any change to demand/the number of projects undertaken? Furthermore, has there been any change in diversity of new projects undertaken post, or in this time frame?
2.  Has there been any change in relationships from any aspect in this time period?
3.  Any other contracting responses which have not been covered?

### Risk management responses

1.  Has there been any change to risk identification/mitigation policy or legislative/regulatory changes regarding risk?
2.  Have you witnessed any differences to the risks associated with projects now which were not present before the period?
3.  Any other risk management responses which have not been which you have witnessed?

### Cost control responses

1.  Has the company conducted any cost reduction strategies to your knowledge, if so, what strategies have they employed?
2.  Have efficacies in any respect changed during this period? (i.e., technological, supply chain, staff, process)
3.  Any other cost control responses which have not been covered?

### Financial-related responses

1.  Have projects been undertaken with little or no profit?
2.  Have there been any diversification/alternate revenue streams exploited during this period?
3.  Any other financial-related responses which have not been covered?

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
