# Peer review of "Response Strategies of UK Construction Contractors to COVID-19 in the Consideration of New Projects"

_buildings, doi:10.3390/buildings12070946_

Round 1

Reviewer 1 Report

This study identifies the response strategies of UK construction companies in the face of the COVID-19 global pandemic. The topic is interesting and there is potential in this paper; however, the paper has several problems that need to be addressed.

1. The title is not suitable. The main research content is about response strategies of UK construction companies. It is better to present it directly in the title.

2. The writing of the paper is not academic enough.

a) For the introduction, the research background should be introduced in a briefer way. Also, the research question should be introduced in a more scientific way.

b) The motivation should be stated in the part of introduction.

c) The research methods should be described more detailedly, including the information of participants.

d) The conclusion should be divided into two paragraphs. One paragraph is about the research findings and significance, and the other one is about limitation and future research.

3. There are some grammar and diction errors throughout the manuscript. The errors need to be corrected.

Author Response

Dear Reviewer,

Thank you for your feedback and comments dated 10 May 2022. We have carefully reviewed the comments and have revised the manuscript accordingly. Our responses are given in a point-by-point manner below.

We hope the revised is now suitable for publication and look forward to hearing from you in due course.

Kind regards,

Ali Rostami

Reviewer

Comments

Response

The title is not suitable. The main research content is about response strategies of UK construction companies. It is better to present it directly in the title.

2. The writing of the paper is not academic enough.

a) For the introduction, the research background should be introduced in a briefer way.

Also, the research question should be introduced in a more scientific way.

b) The motivation should be stated in the part of introduction.

c) The research methods should be described more detailedly, including the information of participants.

d) The conclusion should be divided into two paragraphs. One paragraph is about the research findings and significance, and the other one is about limitation and future research.

3. There are some grammar and diction errors throughout the manuscript. The errors need to be corrected.

Rows 2-5 Title changed to Response strategies of UK construction contractors to COVID19 in the consideration of new projects”

Rows 65-75 omitted.

Rows 151 to 157 now highlights the objectives of the research

Included in introduction

Rows 199 to 219 includes further information.

Row 722 to 738 concludes findings.

Row 739 to 746 makes recommendations for future research.

Entire paper proofread and corrected.

Reviewer 2 Report

The research is timely to study the impact of COVID-19 on the project management and initiatives in the UK construction industry in respect of four aspects – contracting, risk management, cost control and finance. It would be more informative to provide how those four aspects are identified in the research. Are they the concentration areas or key performance indicators in the UK construction industry? How do they relate to each other? The research methodology was based on the qualitative survey of 32 interviews from construction organizations of different size. More information should be given to show how the interviewees were sourced to understand the coverage of the research. It is clear to show the responses by means of tables and categories. More details can be shown on how the tables and classifications can be developed from the interview questions attached to the paper. The research provides comprehensive review of literature to support the responses from the interviewees on the key areas identified and the strategies proposed, and a summary table is required to highlight the recommendations made to enrich the contents of the paper. Kindly consider the comments above so that the paper can meet the required high quality of Buildings for publication.

Author Response

Dear Reviewer,

Thank you for your feedback and comments dated 10 May 2022. We have carefully reviewed the comments and have revised the manuscript accordingly. Our responses are given in a point-by-point manner below.

We hope the revised is now suitable for publication and look forward to hearing from you in due course.

Kind regards,

Ali Rostami

Reviewer

Comments

Response

The research is timely to study the impact of COVID-19 on the project management and initiatives in the UK construction industry in respect of four aspects – contracting, risk management, cost control and finance. It would be more informative to provide how those four aspects are identified in the research.

Are they the concentration areas or key performance indicators in the UK construction industry? 

How do they relate to each other?

 The research methodology was based on the qualitative survey of 32 interviews from construction organizations of different size. More information should be given to show how the interviewees were sourced to understand the coverage of the research.

It is clear to show the responses by means of tables and categories. More details can be shown on how the tables and classifications can be developed from the interview questions attached to the paper.

The research provides comprehensive review of literature to support the responses from the interviewees on the key areas identified and the strategies proposed, and a summary table is required to highlight the recommendations made to enrich the contents of the paper. the required high quality of Buildings for publication.

Included in rows 165 to 175 and table 1.

Included in rows 165 to 175.

Rows 224 to 232 includes sampling.

Row 351 Table 3 added.

Row 447 Table 4 added.

Row 548 Table 5 added.

Row 657 Table 6 added.

Row 93 Table 1 added.
